# Monitored Supplementation of Vitamin D in Preterm Infants: A Randomized Controlled Trial

**DOI:** 10.3390/nu13103442

**Published:** 2021-09-28

**Authors:** Alicja Kołodziejczyk-Nowotarska, Renata Bokiniec, Joanna Seliga-Siwecka

**Affiliations:** Department of Neonatology and Neonatal Intensive Care, Medical University of Warsaw, 00-315 Warsaw, Poland; zuzialicja@gmail.com (A.K.-N.); Renata.bokiniec@wum.edu.pl (R.B.)

**Keywords:** vitamin D, osteopenia, prematurity, metabolic bone disease, rickets

## Abstract

Appropriate supplementation of vitamin D can affect infections, allergy, and mental and behavioral development. This study aimed to assess the effectiveness of monitored vitamin D supplementation in a population of preterm infants. 109 preterm infants (24 0/7–32 6/7 weeks of gestation) were randomized to receive 500 IU vitamin D standard therapy (*n* = 55; approximately 800–1000 IU from combined sources) or monitored therapy (*n* = 54; with an option of dose modification). 25-hydroxyvitamin D [25(OH)D] concentrations were measured at birth, 4 weeks of age, and 35, 40, and 52 ± 2 weeks of post-conceptional age (PCA). Vitamin D supplementation was discontinued in 23% of infants subjected to standard treatment due to increased potentially toxic 25(OH)D concentrations (>90 ng/mL) at 40 weeks of PCA. A significantly higher infants’ percentage in the monitored group had safe vitamin D levels (20–80 ng/mL) at 52 weeks of PCA (*p* = 0.017). We observed increased vitamin D levels and abnormal ultrasound findings in five infants. Biochemical markers of vitamin D toxicity were observed in two patients at 52 weeks of PCA in the control group. Inadequate and excessive amounts of vitamin D can lead to serious health problems. Supplementation with 800–1000 IU of vitamin D prevents deficiency and should be monitored to avoid overdose.

## 1. Introduction

Vitamin D plays an important role in skeletal health [1], and its receptors are in most tissues. Therefore, an appropriate supplementation in perinatal and neonatal periods affects respiratory infections, sepsis, allergy, and mental and behavioral development [2,3,4,5,6]. Neonatal vitamin D storage depends on 50–70% of the maternal 25-hydroxyvitamin D [25(OH)D] levels received by newborns [7]. However, preterm infants are vulnerable to vitamin D deficiency (VDD) because of maternal vitamin D supply deprivation and exposure to additional risk factors, such as long-term parenteral nutrition use, intolerance to human milk fortifiers and formulas, and neonatal cholestasis [8]. In the premature infants’ population, VDD can lead to bone disease, which is described as rickets of prematurity, osteopenia of prematurity, or metabolic bone disease (MBD) [9]. Nevertheless, high vitamin D supplementation among low-birth-weight infants with immature renal filtration can lead to vitamin D toxicity with hypercalcemia or hypercalciuria and cause serious illness [10,11].

The American Academy of Pediatrics (AAP) recommends 200–400 IU/day vitamin D (from all sources) for preterm infants [8]. In contrast, The European Society for Pediatric Gastroenterology Hepatology and Nutrition (ESPGHAN) recommends a higher intake (800–1000 IU/day) during the first months of life to prevent the deficiency caused by prematurity. Thus, higher vitamin D concentrations that can affect extra-skeletal action is implemented [12]. However, the benefits and safety of using higher doses remain controversial [10]. Separate guidelines for Central Europe recommended 400–800 IU/day from the first days of life for preterm infants, followed by 400 IU/day from 40 weeks of post-conceptional age (PCA) [13].

Vitamin D dosage, safety, and effectiveness in preterm infants remain controversial and clear criteria for adequate supplementations have not been established. Some experts have suggested a new approach to vitamin D supplementation-monitored therapy [8,10,14,15]. However, this approach’s clinical significance and long-term effect have not been studied in most preterm populations.

Our trial’s main objective was to compare the effectiveness and safety of monitored vitamin D supplementation with standard 500 IU (800~1000 IU, including enteral nutrition) supplementation in preterm infants born at 24 to 32 weeks of GA. The primary endpoint was defined as an optimum vitamin D status at 40 weeks of PCA. Secondary objectives consisted of the effect of both supplementation modes on 25(OH)D concentrations at 35 and 52 weeks of PCA, MBD prevalence, vitamin D toxicity, and nephrolithiasis.

## 2. Subjects and Methods

### 2.1. Study Design and Participants

We hypothesized that monitored therapy is more effective and safer than the standard therapy in vitamin D-supplemented infants.

This study was a pragmatic, unblinded, parallel-group, and superior randomized controlled trial and the full protocol has been published [16].

Parents and caregivers of eligible infants were invited to participate in the study shortly after birth and admission to the Neonatal Intensive Care Unit at Princess Anne’s Hospital in Warsaw (a tertiary level perinatal center, Department of Neonatology and Neonatal Intensive, Medical University of Warsaw). After providing verbal and written information about the study, an informed consent was obtained from the parents and caregivers of the infants. All procedures were carried out in the Department of Neonatology and Neonatal Intensive Care, Medical University of Warsaw.

We included all preterm infants born between 24–32 weeks of gestation and out-born admitted within 48 h postnatally. At the time of recruitment, caregivers were provided written informed consents and agreed to return for revisits. Very preterm infants (24 0/7–31 6/7) and infants 32 0/7–32 6/7 weeks of gestation (moderate preterm infants, who often require parenteral nutrition) are at higher risk of developing VDD and MBD [8,17]. Thus, the Polish Neonatal Society guidelines recommend vitamin D monitoring in preterm infants born at up to 32 6/7 weeks of gestation [15].

Exclusion criteria were infants born at >32 weeks of gestation with major congenital abnormalities, cholestasis, and severe illness deemed incompatible with survival at birth, when written informed consent was not given, and if there were communication difficulties with caregivers.

Participants were randomized into control and monitored groups within the first 7 d of life after re-evaluating the inclusion and exclusion criteria.

### 2.2. Interventions

Infants of >7 d of age received 500 IU vitamin D (“Devikap” cholecalciferolum, Polpharma, Poland) as per the local protocol. After full enteral feeding was established, vitamin D supplementation consisting of 500 IU and 150–300 IU/kg was included in human milk fortifiers (if fed exclusively with breast milk) or 190 IU/kg in milk formula, depending on the type of feeding (800~1000 IU in total). Blood samples for 25(OH)D concentrations were obtained at 4 weeks of age, and subsequent measurements at 8, 35, 40, and 52 weeks of PCA. Vitamin D doses were appropriately modified based on 25(OH)D concentrations in the monitored group (Figure 1). We hypothesized that most VDD cases in infants are secondary to low maternal status. Thus, we increased vitamin D supplementation by 500 IU in such cases with low vitamin D status at 4 weeks of age, which allowed us to reach higher recommended levels of 1000 IU/day [14]. Blood samples obtained from the randomized infants on standard therapy had the same results as the monitored group but without any changes in dose.

### 2.3. Primary Outcome

The primary endpoint was the number of neonates with deficient or excess vitamin D levels at 40 ± 2 weeks of PCA. Additional 25(OH)D concentrations were performed at birth, 4 and 8 weeks of age, and 35 and 52 ± 2 weeks of PCA. Neonatal nurses collected venous blood samples in glass specimens to assess 25(OH)D concentrations in pre-specified time frames (Figure 1).

An automated quantitative test Vidas^®^ (Biomerieux, Marcy l’Etoile, France) was used to measure 25(OH)D levels. This immune-enzymatic method measures vitamin D2 (25(OH)D2) and vitamin D3 (25(OH)D3) levels in human serum, and characterized by range 8.1–126 ng/mL, <5% coefficient of variation, cross-reactivity 25(OH)D2 91%, and 3-epi-25(OH)D3 2.9% [18,19]. The definitions for vitamin D status slightly differ between local and global health authorities [8,12,13,15,20]. Based on our geographical location and lack of published documents that offer a specific 25(OH)D reference range for preterm infants, we followed the most recent recommendations for Central Europe [13]:Deficiency: 0–20 ng/mL (0–50 nmol/L)Suboptimal concentration: >20–30 ng/mL (>50–75 nmol/L)Optimal concentration: >30–50 ng/mL (>75–125 nmol/L)Increased level: >50–100 ng/mL (125–250 nmol/L)Toxic level: >100–200 ng/mL (>250 nmol/L)Acceptable level 20–80 ng/mL (50–200 nmol/L) *

* Additional level based on safe concentrations described in the most recent regional recommendation [15].

### 2.4. Secondary Outcomes

#### 2.4.1. Osteopenia

The exact definition of osteopenia incidence (metabolic bone disease–MBD) remains unknown due to a lack of consensus on its definition [21]. MBD was defined as decreased bone mineral content relative to the expected level of mineralization for an infant of comparable size or gestational age seen in conjunction with biochemical and ultrasound changes. Neonatal nurses collected venous samples for serum alkaline phosphatase (ALP) and phosphate (P) level analyses at 35, 40, and 52 ± 2 weeks of PCA. Olympus AU 480 (Beckman Coulter, Fullerton, CA, USA) was used for measurements.

Average bone mass (ABM) was assessed using quantitative ultrasound (QUS) Sunlight Premier 7000 (BeamMed, Petah, Israel [22,23,24]. A small ultrasound probe (CRB Probe ROHS, 900–1000 kHz) measures the speed of sound (SOS) in m/s in the axial transmission mode by placing it along the mid tibia. Normal values are undefinable due to high intra-individual variation. However, in a recently published study, preterm infants 24–28 weeks GA) examined at 40 weeks PCA showed significantly lower SOS than term infants [23]. Two previously trained neonatologists not participating in the study and blinded to group allocation assessed ABM in each enrolled patient at 35 and 40 ± 2 weeks of PCA. SOS measurements were made on the mid-tibia shaft and length was determined by calculating the distance from the knee to the heel by placing the probe over the area. Measurements were performed in triplicates, and the mean value was used for the data analysis.

We defined biochemical MBD as serum levels of ALP > 500 IU and *p* < 1.8 mmol/L or ALP > 900 IU [25]. Additionally, we assessed tubular reabsorption of phosphate (TRP), a widely accepted indicator of inadequate phosphate intake, by calculating the phosphorus/creatinine ratio in urine and serum. Phosphate deficiency suppresses parathyroid hormone (PTH) activity and initiates 1,25(OH)D synthesis, leading to increased phosphate reabsorption in the kidney. TRP > 95% with phosphate < 1.8 mmol/L is highly suggestive of osteopenia [21].

#### 2.4.2. Nephrocalcinosis and Nephrolithiasis

Hypercalcemia was defined as serum concentrations of ≥2.75 mmol/L [26], while hypercalciuria was measured by calculating urine calcium/creatinine ratios [27,28], both of which are risk factors for nephrolithiasis in infants.

Ultrasonography has proven good intra-observer reproducibility (kappa 0.84) in preterm infants and is a reliable tool for detecting nephrocalcinosis [29]. A trained ultra-sonographer assessed nephrolithiasis at 35 and 52 ± 2 weeks PCA using Philips HD 11XE (Koninklijke Philips, Eindhoven, Netherlands). Increased medullar echogenicity (small white flecks in the tip of the pyramids) was considered nephrocalcinosis [30].

### 2.5. Adverse Events

All adverse events, defined as any unexpected medical occurrences in a subject without regard to the possibility of a causal relationship, were identified after the subject provided consent and upon study enrolment in the and were recorded after hospital discharge. Those that met the criteria for a serious adverse event (SAE) between study enrolment and hospital discharge were reported to the local Ethical Committee.

### 2.6. Participants’ Retention in the Study

Since most patients in this study faced long-term hospital care, the study team ensured structured staff education and sufficient trial management. All medical records (MR) of included patients were appropriately labeled with bright-colored stickers indicating the study group. A team member audited MRs to schedule appropriate assessments and laboratory samples once a week. Additionally, a list of planned interventions and assessments was given to the attending physician weekly, and we organized a bimonthly departmental meeting to acknowledge any concerns related to the trial. Parents of included infants received verbal and written vitamin D administration instructions and scheduled follow-up appointments upon discharge. Moreover, parents received a reminder text message a day before the scheduled visit.

### 2.7. Data Monitoring

Data monitoring committee was not established since the trial intervention (vitamin D 200–1000 IU) did not differ from the standard of care accepted by several pediatric societies [8,10,12,13,15,31]. The profile of potential side effects was also determined. Since there was potential to exceed safe ranges of vitamin D administration, it was decided to follow up all excluded patients with toxic levels up to 52 weeks of PCA to assess possible side-effects.

### 2.8. Sample Size Calculations

The sample size was calculated based on the main outcome defined as the number of neonates with 25(OH)D deficiency or excess at 40 ± 2 weeks of PCA. Earlier studies reported an incidence of VDD in preterm infants between 64–83% at birth which was reduced by vitamin D supplementation to 40–66% [32,33]. Assuming an estimate prevalence of 83% in the standard supplementation group we chose to detect a decrease by 25% in patients with VDD vs control group with a power of 80% and α = 0.05 to meet acceptable recruitment rates and reach statistically significant results; hence, 57 infants were required in each study group. In addition, we aimed to recruit 138 infants to account for 20% loss to follow-up.

### 2.9. Ethics Approval and Consent to Participate

The Bioethics Committee of the Medical University of Warsaw approved the study (ethics approval and consent #KB143/2014 and amendment #KB10/A/2016). We obtained written informed consent from participants before inclusion.

### 2.10. Statistical Analysis

Statistical analysis was conducted using R software version 3.5.1 (http://cran.r-project.org, accessed on 20 September 2021), while nominal variables were presented with n (%) and continuous variables as mean ± SD. Furthermore, normality of distribution was verified based on Shapiro-Wilk test, skewness, and kurtosis values. The equity of variances between the group was assessed using Levene’s test. Moreover, the groups’ comparison for the primary and secondary outcomes was carried out using the chi-square test or the Fisher exact test for nominal variables and *t*-test for continuous variables. Vitamin D level between both groups was also compared with ANCOVA taking into account following covariates: GA, birth weight, vitamin D supplementation in pregnancy, 25(OH)D at birth. All tests were two-tailed with α = 0.05. Additionally, the mean difference (MD) or relative risk (RR) between groups was calculated, both with a 95% confidence interval (CI).

## 3. Results

Of the 131 eligible preterm infants admitted between May 2017 and December 2019, 109 with a gestational age of 24 weeks, 0 d to 32 weeks, and 6 d were randomly assigned to receive either monitored or standard vitamin D supplementation. A total of 10 infants were excluded in the monitored group and 13 in the standard therapy group (Figure 2).

All patients were reassessed until 40 weeks of PCA. Forty-three and 37 infants respectively randomized to monitored and standard therapy were followed up to 52 weeks of PCA. Additionally, the decision was made to terminate supplementation in 10 (23%) patients of the control group secondary to an extremely high vitamin D status (≥90 ng/mL), and significant risk of further overdosing up to 52 weeks of PCA. These patients were included in the secondary outcome analyses at 52 weeks of PCA (intention to treat analysis). One protocol deviation was registered in the control group, wherein the infant received incorrect vitamin D dosing and was excluded from the analysis. The trail was terminated early because of high percentage of toxic levels of vitamin D in the control group at 40 weeks of PCA.

Baseline characteristics are shown in Table 1.

Mean gestational age was similar in both groups; 29 weeks in the monitored vs. 28 weeks in the control group. There were more extremely preterm infants in the control group than the monitored group (though the difference was not significant), but the mean birth weight was similar in both groups. Maternal pregnancy supplementation and cord blood 25(OH)D levels were similar in both groups. Due to laboratory technical difficulties, 25(OH)D concentrations of a patient in the monitored group at 40 weeks of PCA were not available; thus, 86 patients were included for treatment analysis. Supplementation modification in the monitored therapy is reported below, the control group dose was not modified (500 IU) (Figure 3) (Table 2).

There was no difference in the abnormal status of vitamin D at any of the study points, including 40 w PCA (RR −2,16 95% CI −12,17:7,86). Vitamin D deficiency at 4 weeks of PCA was detected in 18% (8/45) in the monitored vs. 14% (6/42) in the control group (*p* = 0.403) and decreased over time (one case was recorded in the control group at 52 weeks of PCA). Additionally, there was a significant positive correlation between vitamin D status at 4 weeks of PCA with vitamin D supplementation level at pregnancy (ρ = 0.37, *p* < 0.001) and with vitamin D status at delivery (ρ = 0.42, *p* < 0.001). A significantly higher percentage of infants in the monitored group had acceptable vitamin D status (20–80 ng/mL) at 52 weeks of PCA (RR 1.32 95% CI 1.02; 1.71; *p* = 0.017) (Table 3).

Furthermore, at 52 weeks of PCA, most patients presented with an optimal average of 25(OH)D concentration (50.63 ± 18.25 ng/mL) in the monitored group, as opposed to increased concentration registered in the control group (62.19 ± 25.31 ng/mL) (MD −11.56 95% CI −21.96: −1.16, *p* = 0.030) (Figure 4). After considering the following covariates: GA age, birth weight, vitamin D supplementation in pregnancy, 25(OH)D at birth our conclusions remain the same (*p* = 0.023).

There was no difference in the secondary objectives (Table 4).

However, biochemical markers of vitamin D toxicity were observed in two patients at 52 weeks of PCA in the control group with a notable coincidence of increased vitamin D status and abnormal ultrasound findings in five infants.

## 4. Discussion

To the best of our knowledge, we are the first to report a novel approach to monitored vitamin D supplementation among preterm infants, consistent with European guidelines. Our results suggest that implementing monitored vitamin D supplementation does not significantly decrease the incidence of abnormal vitamin D levels at 40 weeks of PCA in a population of preterm infants. However, patients with vitamin D deficiency systematically decreased over time and were comparable in monitored and control groups. Additionally, vitamin D deficiency at 4 weeks of age was positively correlated with lack of maternal supplementation during pregnancy. The MBD incidence did not differ between groups. Bone mineral status was similar, showing decreased levels compared to term infants in both groups. The average 25(OH)D concentration was within the optimal range at 4 weeks of age and 35 weeks of PCA, slightly increased at 40 weeks of PCA in both groups, and further increased at 52 weeks of PCA in the control group. Moreover, achieving acceptable levels at 52 weeks of PCA was persistent in the monitored group. During our observation period, the excessive levels of vitamin D incidence were more frequent in patients with the standard approach. Vitamin D supplementation was discontinued in nearly 25% of infants allocated to standard treatment due to extremely high vitamin D levels (≥90 ng/mL) at 40 weeks of PCA. We observed biochemical features of vitamin D toxicity in two cases and nephrolithiasis associated with very high vitamin D levels in five cases.

Eighty percent of calcium and phosphorus placental transfer occurs between 24–40 weeks of gestation; hence, preterm infants are especially prone to adverse effects of VDD, including MBD and low bone mineral status [21,34,35]. However, high-dose supplementation without monitoring can lead to overdosing despite the lack of consensus in vitamin D toxicity threshold (100 or 150 ng/mL) [10,11,31]. Some experts suggest lower levels, such as 75–80 ng/mL [11,15], due to the increased risk of hypercalcemia in preterm infants [15,36]. Our results provide important information on why preterm infants present with isolated hypercalciuria, which can lead to nephrocalcinosis aggravated by high vitamin D levels [30,37] and a relatively long 25(OH)D half-life [11]. Monitored supplementation might be the best choice to obtain the benefits of vitamin D while avoiding its toxic side effects.

Prospective observational studies have assessed recent recommendations for vitamin D supplementation in preterm neonates. Cho et al. [38] published a study evaluating 25(OH)D concentration at 36 weeks of post menstrual age (PMA) in 43 very low birth weight (VLBW) infants receiving early vitamin D supplementation of 800 IU (total cumulative dose did not exceed 900 IU/day). VDD at 36 weeks of PMA was not recorded, insufficient levels were noted in 21% of infants and excessive levels in three patients. Nephrocalcinosis related to excessive vitamin D status was observed in one patient. A study conducted by Matejek et al. [39] described vitamin D status in 81 VLBW infants with a total vitamin D intake of 800–1000 IU. In this study, 59% of preterm newborns at 37 ± 2 weeks of PMA still presented with VDD. Similar to Cho, we observed a smaller percentage of VDD and insufficiency at 35 weeks of PMA in infants allocated to standard treatment. These findings are different from the report by Matejek, where VDD persisted in more than half of the infants receiving standard management. Notably, our observation period for the potential overdosing period was longer than other studies. This is consistent with Abrams’ latest review, in which the author suggests that potential toxic levels of vitamin D may present in later infanthood [10]. In contrast, Monangi et al. [32] evaluated low-dose supplementation in 120 preterm infants born <32 weeks of GA at 36 weeks of PMA or at discharge. Daily vitamin D intake from vitamin D drops was 200 and 400 IU at discharge from all sources. Thirty-six percent of infants at 36 weeks of PMA or upon discharge still presented with VDD. The authors concluded that this approach led to suboptimal vitamin D intake. A study conducted by McCarthy et al. [40] evaluated the total prolonged, approximately 400 IU/day of vitamin D intake among preterm infants with VDD assessment at 3 weeks of chronological age. The authors identified that 13% of 148 preterm infants born below 33 weeks of GA and VLBW still suffered from VDD, and 8% presented with increased vitamin D concentrations (50–80 ng/mL). The authors concluded that low-dose supplementation in a long-term observation compared to high-dose supplementation did not prevent VDD in most preterm infants. However, high dose supplementation could lead to excessive vitamin D levels if implemented in clinical practice.

The different dosing regimens were compared in randomized trials. Natarajan et al. [33] enrolled 96 preterm infants (28–32 weeks of GA) into two groups of 400 and 800 IU/day of total vitamin D intake. The primary outcome was VDD at 40 weeks of PMA and the secondary outcomes included VDD, bone mineral density (BMD), and biochemical markers of vitamin D status at 3 months of CA. VDD prevalence in the 800 IU/day group was significantly lower than that in the 400 IU/day group at 40 weeks of PCA (38% vs. 67%, RR 0.57, 95% CI 0.14–0.90). One infant in the 800 IU group had excessive vitamin D of 141 ng/mL at 3 months of CA without hypercalcemia, hypercalciuria, and nephrolithiasis. BMD and biochemical markers of vitamin D status did not differ between the groups. Therefore, the authors concluded that a high dose supplementation reduces VDD prevalence but with possible overdosing risk. Fort et al. [41] randomized 100 infants 23–28 weeks of GA) to groups of patients receiving placebo of 200 and 800 IU (providing approximating 200, 400, and 1000 IU total intake). At postnatal day 28, the 800 IU group had a significantly higher average of 25(OH)D levels (22 ng/mL vs. 39 ng/mL vs. 85 ng/mL, *p* < 0.05), resulting in vitamin D overdosing, which was observed sooner than in other studies. Unfortunately, the authors did not examine the clinical features of vitamin D toxicity. Based on their results, Fort et al. suggested a new strategy by administering an initial short high supplementation followed by a lower dose to prevent overdosing. In a double-blind controlled trial by Tergestina et al. [42], vitamin D supplementation of 1000 vs. 400 IU in 99 preterm infants was analyzed (27–34 weeks of GA). At 40 weeks of PMA, the 1000 IU group had a significantly higher average of 25(OH)D levels (47 ng/mL vs. 17 ng/mL, *p* < 0.001). As part of the secondary outcomes, the group reported significantly elevated mean parathormone levels in the 400 IU group compared to the 1000 IU group (*p* = 0.007). An increased vitamin D status (>70 ng/mL) in 9.8% of the 1000 IU group with evidence of hypercalciuria in two out of five patients was observed. However, these results must be interpreted cautiously because the supplementation’s potential effect was based on a single measurement.

According to the aforementioned studies, inappropriate vitamin D supplementation may lead to VDD and overdosing with mild hypercalcemia or hypercalciuria. To the best of our knowledge, we are the first to report a randomized study evaluating monitored supplementation using 25(OH)D concentrations, which can optimize vitamin D supplementation. This approach allows the evaluation of short-term and long-term changes of vitamin D status, including bone metabolism benefits and toxicity risk.

However, our study has some limitations. First, due to the nature of intervention, we were unable to blind the lead physician. Second, the calculated sample size was not achieved due to ethical reasons. We decided to terminate the trial early due to reported cases of vitamin D toxicity levels. Finally, 25(OH)D concentration was measured using an immunoenzymatic method with limitations of overestimating vitamin D status’ biological potential due to cross-reactivity to epimer or underestimation due to non-reactivity to the active D2 form. However, a cross-reactivity to epimer described in the VIDAS method is residual, and the supplements from drops and additional sources in our study contain only the D3 form. Unfortunately, while choosing secondary outcomes, we did not include parathormone levels, which might be a sensitive indicator of calcium-phosphate balance based on recently published research [17,42,43,44]. However, we included TRP and urinary calcium-creatinine ratios, which are sensitive markers of sufficient bone metabolism.

## 5. Conclusions

Our findings suggest that European recommendations of high dose vitamin D supplementation in preterm infants should be implemented under 25(OH)D monitoring to prevent overdosing and renal diseases. A future, well designed RCT with adequate samples is required to confirm that an individualized approach could be beneficial. Furthermore the trial should include a third arm of low-dose US standard recommendation and supplementation to evaluate all available regimes.

## Figures and Tables

**Figure 1 nutrients-13-03442-f001:**
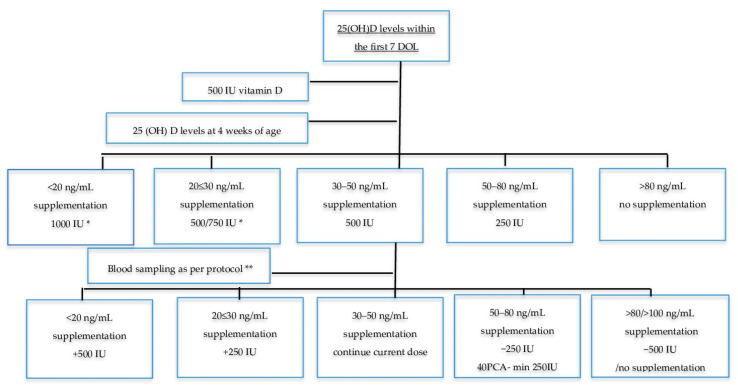
Monitored therapy schedule. DOL: days of life; PCA: post conceptional age; GA: gestational age; * 500 IU for infants < 1.5 kg, 750 IU for infants ≥ 1.5 kg; ** 8 weeks of age, 3 weeks for infants born < 26 GA, 35 weeks PCA ± 2 weeks, 40 ± 2 weeks PCA, 52 +/− 2 weeks PCA.

**Figure 2 nutrients-13-03442-f002:**
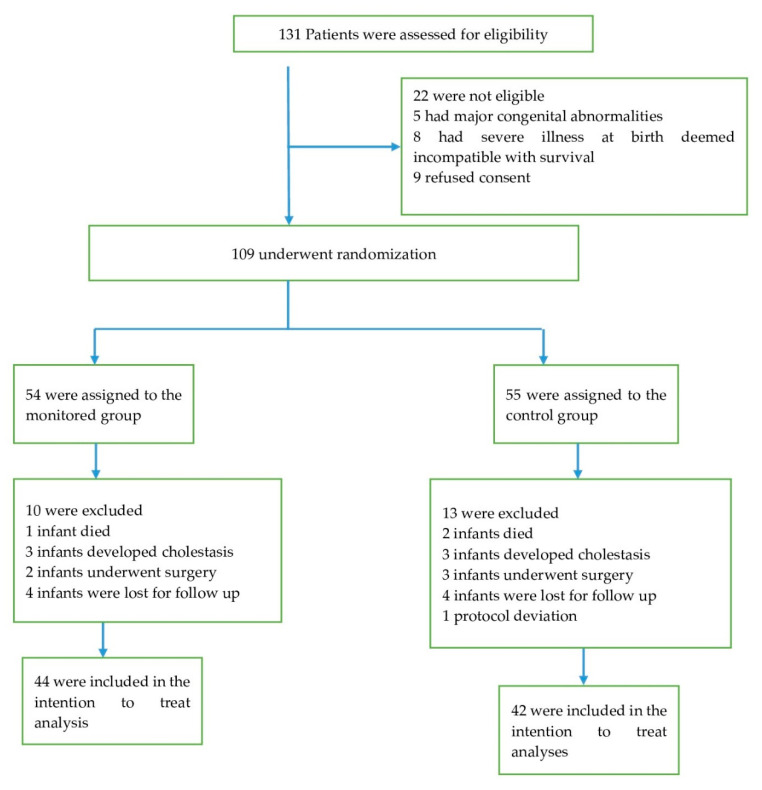
Patient recruitment characteristics.

**Figure 3 nutrients-13-03442-f003:**
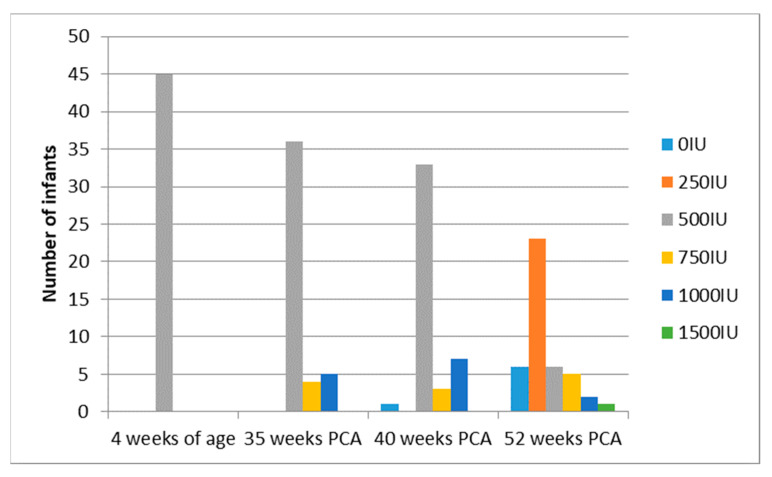
Scheme modified vitamin D supplementation in the monitored group.

**Figure 4 nutrients-13-03442-f004:**
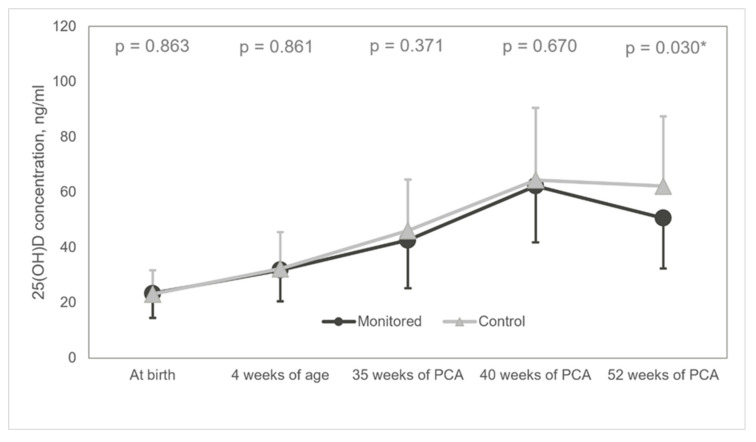
Diagram of mean 25(OH)D concentrations. 25(OH)D: 25-hydroxyvitamin D; * statistically significant.

**Table 1 nutrients-13-03442-t001:** Baseline characteristics of the study groups.

	Monitored Group		Control Group		*p*
	(N = 45)		(N = 42)		
Sex, female, n (%)	45	17 (37.8)	42	19 (45.2)	0.625
GA, weeks	45	29.22 ± 2.16	42	28.76 ± 2.27	0.171
GA < 28 weeks, n (%)	45	9 (20.0)	42	17 (40.5)	0.064
Mean birth weight (g)	45	1380.89 ± 431.14	42	1313.81 ± 471.87	0.490
Birth weight < 1000 g, n (%)	45	8 (17.8)	42	13 (31.0)	0.236
Birth weight condition, n (%) ^1^					
SGA	45	4 (8.9)	42	4 (9.5)	>0.999
Eutrophy		38 (84.4)		35 (83.3)	
Hypertrophy		3 (6.7)		3 (7.1)	
Weight upon discharge, n (%) ^1^					
Hypotrophy	45	18 (40.0)	42	20 (47.6)	0.448
Eutrophy		27 (60.0)		21 (50.0)	
Hypertrophy		0 (0.0)		1 (2.4)	
Additional source of vitamin D, n (%) ^1^					
HMF	45	31 (68.9)	42	26 (61.9)	0.216
HMF/hydrolyzed infant formula		0 (0.0)		2 (4.8)	
HMF/infant formula		9 (20.0)		5 (11.9)	
Infant formula		5 (11.1)		9 (21.4)	
Dose of vitamin D supplementation in pregnancy, IU	45	1031.11 ± 888.77	42	1214.29 ± 870.82	0.335
No vitamin D supplementation in pregnancy, n (%)	45	12 (26.7)	42	8 (19.0)	0.556
Vitamin D supplementation in pregnancy ≥ 2000 IU, n (%)	45	17 (37.8)	42	19 (45.2)	0.625
25(OH)D at birth, ng/mL	43	23.40 ± 8.90	38	23.13 ± 8.59	0.893
Vitamin D at birth, n (%)					
Deficiency	43	14 (32.6)	38	11 (28.9)	0.738
Suboptimal		19 (44.2)		20 (52.6)	
Optimal		10 (23.3)		7 (18.4)	

Data presented as mean ± SD unless otherwise indicated. Groups compared with chi-square test or Fisher exact test ^1^ for nominal variables and with independent samples *t*-test for continuous variables. 25(OH)D: 25-hydroxyvitamin D; GA: gestational age; SGA: small for gestational age; HMF: human milk fortifier.

**Table 2 nutrients-13-03442-t002:** Changes in Vitamin D supplementation and their influence on serum concentration in the monitored group.

Vitamin D Supplementation Change	Vitamin D Concentration (ng/mL)		
	n	Mean	SD
Baseline level	baseline		
500 IU	45	27.18	10.34
Change (baseline-35 weeks of PCA)	35 weeks of PCA		
No change	36	44.31	18.27
Plus 250 IU	4	35.25	10.90
Plus 500 IU	5	36.60	14.59
Change (35 weeks of PCA-40 weeks of PCA)	40 weeks of PCA		
Minus 500 IU	3	72.00	3.46
Minus 250 IU	11	64.00	16.32
No change	23	68.17	19.33
Plus 250 IU	2	38.00	1.41
Plus 500 IU	5	35.20	15.97
Change (40 weeks of PCA-52 weeks of PCA)	52 weeks of PCA		
Minus 1000 IU	1	27.00	n/a
Minus 500 IU	5	44.00	12.71
Minus 250 IU	19	56.16	20.59
No change	14	50.43	16.62
Plus 250 IU	3	43.67	1.53
Plus 500 IU	1	26.00	n/a

Data presented as mean ± SD; PCA: post conceptional age.

**Table 3 nutrients-13-03442-t003:** Primary outcomes 25(OH)D levels between groups.

	Monitored Group		Standard Group		MD/RR (95% CI)	*p* (*p* adj 3)
	N		N			
40 weeks of PCA						
25(OH)D concentration, (ng/mL ^1^)	44	62.27 ± 20.40	42	64.43 ± 26.09	−2.16 (−12.17; 7.86)	0.670 (0.842)
Vitamin D deficiency or excess	44	3 (6.8)	42	6 (14.3)	0.48 (0.13; 1.79)	0.308
Vitamin D deficiency	44	1 (2.3)	42	1 (2.4)	n/a	0.363
Vitamin D suboptimal		1 (2.3)		4 (9.5)		
Vitamin D optimal		9 (20.5)		9 (21.4)		
Vitamin D increased		31 (70.5)		23 (54.8)		
Vitamin D excess		2 (4.5)		5 (11.9)		
Vitamin D acceptable ^2^	44	35 (79.5)	42	29 (69.0)	1.15 (0.90; 1.48)	0.385
25(OH)D > 90 ng/mL	44	2 (4.5)	42	10 (23.8)	0.19 (0.04;0.82)	0.013
4 weeks of PCA						
25(OH)D concentration, ng/mL ^1^	45	31.87 ± 11.39	42	32.33 ± 13.25	−0.46 (−5.76;4.82)	0.861 (0.840)
Vitamin D deficiency or excess ^2^	45	8 (17.8)	42	6 (14.3)	1.24 (0.47; 3.29)	0.774
Vitamin D deficiency	45	8 (17.8)	42	6 (14.3)	n/a	0.827
Vitamin D suboptimal		13 (28.9)		15 (35.7)		
Vitamin D optimal		22 (48.9)		18 (42.9)		
Vitamin D increased		2 (4.4)		3 (7.1)		
Vitamin D excess		-		-		
Vitamin D acceptable ^2^	45	37 (82.2)	42	36 (85.7)	0.96 (0.80; 1.15)	0.774
35 weeks of PCA						
25(OH)D concentration, ng/mL ^1^	45	42.64 ± 17.44	42	46.12 ± 18.43	−3.48 (−11.17; 4.22)	0.371 (0.989)
Vitamin D deficiency or excess	45	5 (11.1)	42	2 (4.7)	2.28 (0.47; 11.11)	0.437
Vitamin D deficiency	45	5 (11.1)	42	2 (4.7)	n/a	0.778
Vitamin D suboptimal		6 (13.3)		5 (11.9)		
Vitamin D optimal		21 (46.7)		20 (47.6)		
Vitamin D increased		13 (28.9)		14 (33.3)		
Vitamin D excess		-		-		
Vitamin D acceptable ^2^	45	39 (86.7)	42	37 (90.2)	0.96 (0.82; 1.12)	0.741
52 weeks of PCA						
25(OH)D concentration, ng/mL ^1^	43	50.63 ± 18.25	27	62.19 ± 25.31	−11.56 (−21.96; −1.16)	0.030 (0.023)
Vitamin D deficiency or excess	43	1 (2.3)	27	4 (14.8)	0.16 (0.02; 1.33)	0.069
Vitamin D deficiency	43	-	27	1 (3.7)	n/a	0.161
Vitamin D suboptimal		4 (9.3)		-		
Vitamin D optimal		20 (46.5)		12 (44.4)		
Vitamin D increased		18 (41.9)		11 (40.7)		
Vitamin D excess		1 (2.3)		3 (11.1)		
Vitamin D acceptable ^2^	43	40 (93.0)	27	19 (70.4)	1.32 (1.02; 1.71)	0.017

Vitamin D: deficiency (25(OH)D < 20 ng/mL), suboptimal (20–30 ng/mL), optimal (30–50 ng/mL), increased (50–100 ng/mL), excess (>100 ng/mL), acceptable (20–80 ng/mL). Data are presented as mean ± SD ^1^ or as n (% of the group) for the remaining variables. Groups were compared using *t*-test ^1^, chi-square test ^2^, or the Fisher exact test (remaining variables). ^3^ Vitamin D level between groups compared without any covariates (*t*-test) and with adjustment to: GA age, birth weight, vit. D supplementation in pregnancy, 25(OH)D at birth (ANCOVA). MD ^1^: mean difference between groups calculated as monitored group minus control group with 95% confidence interval (CI). RR: relative risk between groups calculated as monitored group vs. control group with 95% CI. 25(OH)D: 25-hydroxyvitamin D; PCA: postconceptional age.

**Table 4 nutrients-13-03442-t004:** Secondary outcomes between groups.

	Monitored Group		Standard Group		MD/RR (95% CI)	*p*
	n		n			
MBD, n (%)						
35 weeks PCA	45	7 (15.6)	42	4 (9.5)	1.63 (0.51; 5.18)	0.524
40 weeks PCA	45	2 (4.4)	42	5 (11.9)	0.37 (0.08; 1.82)	0.255
52 weeks PCA	40	1 (2.5)	35	2 (5.7)	0.44 (0.04; 4.62)	0.596
Osteopenia with vitamin D deficiency, n (%)	45	0 (0.0)	42	1 (2.4)	0.31 (0.01; 7.44)	0.483
Hypercalciuria n (%)						
35 weeks PCA	45	7 (15.6)	42	3 (7.1)	2.18 (0.60; 7.88)	0.317
40 weeks PCA ^2^	45	16 (35.5)	42	8 (19.0)	1.87 (0.89;3.90)	0.098
52 weeks PCA	40	4 (10.0)	35	5 (14.2)	0.70 (0.20; 2.40)	0.726
Hypocalcemia, n (%)						
35 weeks PCA	45	2 (4.4)	42	2 (4.8)	0.93 (0.14; 6.33)	>0.999
40 weeks PCA	45	1 (2.2)	42	0 (0.0)	2.80 (0.12; 67.00)	>0.999
52 weeks PCA	40	0 (0.0)	35	0 (0.0)	0.88 (0.02; 43.13)	>0.999
Hypercalcemia, n (%)						
35 weeks PCA	45	1 (2.5)	42	0 (0.0)	2.80 (0.12; 67.00)	>0.999
40 weeks PCA	45	0 (0.0)	42	0 (0.0)	0.66 (0.01; 32.20)	>0.999
52 weeks PCA	40	2 (20.0)	35	1 (2.8)	1.75 (0.17; 18.48)	>0.999
TRP <85, n (%)						
35 weeks PCA	45	1 (2.2)	42	2 (4.8)	0.47 (0.04; 4.96)	0.608
40 weeks PCA	45	0 (0.0)	42	1 (2.4)	0.31 (0.01; 7.44)	0.483
52 weeks PCA	40	0 (0.0)	35	1 (2.8)	0.29 (0.01; 6.96)	0.447
TRP > 95, n (%)						
35 weeks PCA ^2^	45	33 (73.3)	42	30 (71.4)	1.03 (0.79; 1.33)	>0.999
40 weeks PCA ^2^	45	39 (88.6)	42	35 (83.3)	1.04 (0.87; 1.24)	0.767
52 weeks PCA	40	36 (90.0)	35	32 (91.4)	0.98 (0.85; 1.14)	>0.999
Nephrolithiasis, n (%)						
35 weeks PCA	42	0 (0.0)	40	2 (5.0)	0.19 (0.01; 3.85)	0.235
52 weeks PCA	36	3 (8.3)	33	5 (15.2)	0.55 (0.14; 2.12)	0.466
Nephrolithiasis at 52 weeks with vitamin D > 80 at 40 or 52 weeks PCA, n (%)	36	1 (2.8)	33	4 (12.1)	0.23 (0.03; 1.95)	0.186
Bone mass, SOS, m/s ^1^						
35 weeks PCA	24	2 786.33 ± 152.52	26	2 745.88 ± 112.15	40.45 (−35.26; 116.16)	0.288
40 weeks PCA	20	2 751.50 ± 177.52	20	2 743.35 ± 112.27	8.15 (−86.93; 103.23)	0.863
Bone mass Z score ^1^						
35 weeks PCA	24	−1.68 ± 1.24	26	−1.96 ± 1.01	0.28 (−0.35; 0.93)	0.370
40 weeks PCA	20	−3.02 ± 1.49	20	−3.10 ± 0.99	0.08 (−0.74;0.89)	0.853

Data are presented as mean ± SD for bone mass and as n (%) for the remaining variables. Groups are compared using *t*-test ^1^, chi-square test ^2^, or the Fisher exact test (remaining variables). MD ^1^: mean difference between groups calculated as monitored group minus control group with 95% confidence interval (CI). RR: relative risk between groups calculated as monitored group vs. control group with 95% CI. MBD: metabolic bone disease; TRP: tubular reabsorption phosphate PCA: post conceptional age.

## Data Availability

Datasets generated and analyzed during the current study are available upon reasonable request from the corresponding author.

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
