# Peer review of "Monitored Supplementation of Vitamin D in Preterm Infants: A Randomized Controlled Trial"

_nutrients, 2021, doi:10.3390/nu13103442_

Round 1
Reviewer 1 Report
Summary
The authors hypothesised that a program of monitored administration of Vitamin D, compared with a standard regime, would be clinically effective and safe in preterm infants. To answer this question, they designed a pragmatic randomised trial in preterm infants born between 24-32 weeks of gestation.
Positive Comments
- Well-structured background information and development of a clinical question.
- Acknowledgement of uncertainties in the current knowledge, with a clear explanation on how these were dealt with pragmatically in the protocol.
- Detailed description of methods with consideration for a thorough assessment of efficacy and safety (apart from PTH assessment).
- I would like to commend the authors on the hypothesis and design of the study. The reporting of results (as detailed below) can improve the overall quality of the manuscript.
Minor Points
- The authors have stated that a Data Monitoring Committee (DMC) was not set up since “the trial intervention (vitamin D 200-1000 IU) did not differ from the standard of care accepted by several paediatric societies.” However, in the intervention group, there was potential to exceed this range of Vitamin D supplementation based on serum concentrations; indeed, this was the hypothesis of the study. As a result, a DMC should have been considered even though the authors have monitored all patients for any potential side-effects. The reviewer is fully aware that this cannot be undone now but suggest considering adding a statement to this effect.
- The incidence of metabolic bone disease (MBD) in preterm infants, like other morbidities in this population, is inversely proportional to gestation. The more preterm an infant is, the higher the risk of developing MBD. Conversely, moderate and late preterm infants have a much smaller risk of developing MBD. The authors have chosen to include infants born up to 33+6 weeks of gestation- who are less likely to benefit from the intervention but would still be exposed to the risks of the intervention. Please consider justifying the choice of gestations along with observed incidence at various gestations.
- A final “Conclusions” section should be considered.
Major Issues
- The sample size estimation needs more detail The authors have stated that they wanted to see a reduction of VDD of 25%, the baseline incidence is not stated. This is crucial in determining the estimate.
- I have concerns regarding the exclusion of 10 infants in the standard group, who had high serum concentration of Vitamin D, from secondary outcomes analysis. I fully agree with stopping further supplementation in these infants; however, the authors need to explain why they were not assessed for secondary outcomes at 52 weeks.
- The actual amount of Vitamin D supplementation needs to be reported.
- The changes in Vitamin D supplementation with serum concentration in the monitored group needs to be reported.
Author Response
Dear Reviewer,
Thank you for your review of our paper. We have answered each of your points below.
Minor Points
- The authors have stated that a Data Monitoring Committee (DMC) was not set up since “the trial intervention (vitamin D 200-1000 IU) did not differ from the standard of care accepted by several paediatric societies.” However, in the intervention group, there was potential to exceed this range of Vitamin D supplementation based on serum concentrations; indeed, this was the hypothesis of the study. As a result, a DMC should have been considered even though the authors have monitored all patients for any potential side-effects. The reviewer is fully aware that this cannot be undone now but suggest considering adding a statement to this effect.
We have included a sufficient statement in the paper.
- The incidence of metabolic bone disease (MBD) in preterm infants, like other morbidities in this population, is inversely proportional to gestation. The more preterm an infant is, the higher the risk of developing MBD. Conversely, moderate, and late preterm infants have a much smaller risk of developing MBD. The authors have chosen to include infants born up to 33+6 weeks of gestation- who are less likely to benefit from the intervention but would still be exposed to the risks of the intervention. Please consider justifying the choice of gestations along with observed incidence at various gestations.
We have clarified our choice of patients together with the incidence of MBD at various gestations.
- A final “Conclusions” section should be considered.
Due to an editorial mistake, in our previous manuscript the conclusion section was not preceded with a clear title. We have made the appropriate changes.
Major Issues
- The sample size estimation needs more detail The authors have stated that they wanted to see a reduction of VDD of 25%, the baseline incidence is not stated. This is crucial in determining the estimate.
We have provided a more detailed description of the sample size estimation.
- I have concerns regarding the exclusion of 10 infants in the standard group, who had high serum concentration of Vitamin D, from secondary outcomes analysis. I fully agree with stopping further supplementation in these infants; however, the authors need to explain why they were not assessed for secondary outcomes at 52 weeks.
We have clarified this paragraph and included these patient to analysis using intention to treat analysis a more detailed description of the secondary outcome analysis. Thank you for this important comment.
- The actual amount of Vitamin D supplementation needs to be reported.
We have provided additional information in Figure 3, presenting the true amount of vitamin D supplementation in the monitored group.
- The changes in Vitamin D supplementation with serum concentration in the monitored group needs to be reported.
We have provided additional information in Table 2, on the changes of vitamin D supplementation together with 25(OH)D concentration in the monitored group.
Reviewer 2 Report
Authors performed a randomized controlled trial aiming to assess the effectiveness of monitored vitamin D supplementation in a population of preterm infants, up to 52 weeks of postconceptional age.
The study has important limitations (i.e. sample power not reached for ethical reasons, the use of immunoenzymatic method that overestimate vitamin D status) well described in discussion section. Despite this, I have some concerns in a major revision, that might improve the manuscript for the publication in this prestigious Journal.
Methods:
- I suggest to unify paragraph 2.1 to 2.6 in a sole paragraph entitled “Study design and participants”
- Lines 170-173. Despite the exact definition of osteopenia remains unknown, a reference should be added for the definition used in the present work or Authors should declare that it is an arbitrary definition.
- I suggest to add the p value in Table 1 and to perform multivariate analysis to adjust the outcome for the baseline characteristics statistically different between the two groups (in case of difference statistically significant) and for variables that, basing on literature research, could influence the outcome.
Results:
- Figure 2. In control group “13 were excluded: 2 infants died, 3 infants developed cholestasis, 3 infants underwent surgery, 4 infants were lost for follow up”. Are they 12 or one is missing?
- Figure 3, p values are missing. I believe that the message in a figure must be immediate. For the variables statistically significant, the p values should be underlined also in the figure not only in the text.
- Each table should have a table legend with the description of all the abbreviations cited in the text of the table.
Discussion:
- Lines 352-353. Has been at term newborns enrolled in this study or this is a bibliography reference? In the second case reference is missing.
- In conclusion section, I suggest to underline that other RCTs with less methodological limitations and adequate sample size should confirm these results.
General comment:
- Please modify references style as suggested by the Journal guidelines (https://www.mdpi.com/journal/nutrients/instructions). I suggest to modify the references with a bibliography software package, such as EndNote, ReferenceManager or Zotero.
- At the end of the article there is a typing mistake with “conclusion” section.
- Please, correct and complete affiliation, mails and correspondence as Journal guidelines requested (https://www.mdpi.com/journal/nutrients/instructions)
- I suggest an important revision by a native English speaker to improve the readability of the manuscript. The Journal offers English editing service (https://www.mdpi.com/authors/english).
Author Response
Dear Reviewer,
Thank you for your review of our paper. We have answered each of your points below.
Methods:
- I suggest to unify paragraph 2.1 to 2.6 in a sole paragraph entitled “Study design and participants”
As suggested, we have made the appropriate adjustments.
- Lines 170-173. Despite the exact definition of osteopenia remains unknown, a reference should be added for the definition used in the present work or Authors should declare that it is an arbitrary definition.
We have provided a reference for osteopenia.
- I suggest to add the p value in Table 1 and to perform multivariate analysis to adjust the outcome for the baseline characteristics statistically different between the two groups (in case of difference statistically significant) and for variables that, basing on literature research, could influence the outcome.
As requested, we have added the p value n table 1, and a multivariate analysis was performed and reported.
Results:
- Figure 2. In control group “13 were excluded: 2 infants died, 3 infants developed cholestasis, 3 infants underwent surgery, 4 infants were lost for follow up”. Are they 12 or one is missing?
This was a typographical error, which has been resolved – 1 infant was excluded because due to protocol deviation.
- Figure 3, p values are missing. I believe that the message in a figure must be immediate. For the variables statistically significant, the p values should be underlined also in the figure not only in the text.
We have provided p values in Figure 3.
- Each table should have a table legend with the description of all the abbreviations cited in the text of the table.
We have provided descriptions of all abbreviations cited in the text of the table.
Discussion:
- Lines 352-353. Has been at term newborns enrolled in this study or this is a bibliography reference? In the second case reference is missing.
The phrase “post menstrual age” has been added to the text, which clarify this sentence.
- In conclusion section, I suggest to underline that other RCTs with less methodological limitations and adequate sample size should confirm these results.
We have added the suggested phrase.
General comment:
- Please modify references style as suggested by the Journal guidelines (https://www.mdpi.com/journal/nutrients/instructions). I suggest to modify the references with a bibliography software package, such as EndNote, ReferenceManager or Zotero.
While preparing our manuscript, we used the Endnote software package, however due to technical issues, the reference was not edited correctly. We have made the appropriate changes.
- At the end of the article there is a typing mistake with “conclusion” section.
We have made the appropriate changes.
- Please, correct and complete affiliation, mails and correspondence as Journal guidelines requested (https://www.mdpi.com/journal/nutrients/instructions)
We have made the appropriate changes.
- I suggest an important revision by a native English speaker to improve the readability of the manuscript. The Journal offers English editing service (https://www.mdpi.com/authors/english).
As stated in the acknowledgment section, we submitted our manuscript to Editage, an editing service company recommended by the BMJ Journals group (https://www.editage.com). The manuscript was edited and proofread by a native English speaker. However, we are happy to resubmit the manuscript for final editing by Editage if required.
Round 2
Reviewer 2 Report
- Statistical section must be improved with the multivariate analysis reported in the text after the revision.
- I suggest to add the confirm of the results by multivariate analysis also in the abstract section.
- Please keep attention, the affiliations and mails of all authors must be provided as requested by the Journal (https://www.mdpi.com/journal/nutrients/instructions)
Author Response
。